# Occurrence of *Eimeria* spp. and Intestinal Helminths in Free-Range Chickens from Northwest and Central Romania

**DOI:** 10.3390/ani14040563

**Published:** 2024-02-07

**Authors:** Mircea Coroian, Tünde-Zsuzsánna Fábián-Ravasz, Patricia Roxana Dobrin, Adriana Györke

**Affiliations:** 1Department of Poultry Management and Pathology, University of Agricultural Sciences and Veterinary Medicine of Cluj-Napoca, 400372 Cluj-Napoca, Romania; 2Department of Parasitology and Parasitic Diseases, University of Agricultural Sciences and Veterinary Medicine of Cluj-Napoca, 400372 Cluj-Napoca, Romania; rtundike@yahoo.com (T.-Z.F.-R.); dobrin.patricia98@gmail.com (P.R.D.); adriana.gyorke@usamvcluj.ro (A.G.)

**Keywords:** endoparasites, chicken, free-range, *Eimeria*, *Ascaridia*, *Heterakis*, *Capillaria*, *Strongyle*

## Abstract

**Simple Summary:**

In the context of backyard poultry raising, a notable concern arises regarding the susceptibility to parasitic infections. The poultry industry holds a crucial position in ensuring food safety and nutritional requirements, emerging as the most rapidly advancing agricultural sub-sector. The aim of our study was to assess the prevalence of gastrointestinal parasites in chickens raised in the backyard system within the northwestern and central regions of Romania. Fecal samples were collected and tested using flotation, McMaster, and PCR (polymerase chain reaction) methods. The overall prevalence of infection with gastrointestinal parasites was 53.1%. Intestinal parasites demonstrate a pronounced prevalence within the context of backyard poultry flocks, and the substantial burdens imposed by these parasites can deleteriously influence both avian productivity and economic considerations.

**Abstract:**

Chickens raised in backyard free-range systems are confronted with a significant threat of parasitic infections. Among the parasitic agents, protozoa belonging to the genus *Eimeria* and helminths, including *Ascaridia galli*, *Capillaria* spp., *Heterakis gallinarum*, and *Strongyloides avium*, stand out as the most prevalent. The sampling protocol included sixteen localities in four counties within the Transylvania region of Romania. Fecal samples were collected from chickens reared in a backyard system. Fecal samples were screened for oocysts (O) and eggs (E) by flotation method, and their number per gram of feces (OPG/EPG) was calculated after counting them by McMaster method. Positive samples for *Eimeria* spp. were further analyzed by PCR (polymerase chain reaction) method to identify the *Eimeria* species. A total of 145 flocks were tested and the overall prevalence of infection was 53.1%. The most prevalent infections were with *A. galli/H. gallinarum* (25.5%), and *Eimeria* spp. (24.8%), followed by *Capillaria* spp. and strongyles. The mean OPG/EPG values were as follows: 63,577 for *Eimeria* spp., 157 for *Ascaridia/Heterakis*, 362 for *Capillaria* spp., and 1671 for *Strongyle* eggs. Identified *Eimeria* species were *E. acervulina* (41.7%), *E. tenella* (27.8%), *E. praecox* (16.7%), *E. brunetti* (16.7%), OTUy (operational taxonomic unit y) (8.3%), OTUz (operational taxonomic unit z) (8.3%) and *E. mitis* (5.6%). Intestinal parasites exhibit a high prevalence among chickens in backyard poultry flocks, and the presence of significant parasite burdens can adversely affect both productive and economic aspects. To the best of our knowledge, this is the first comprehensive study that aimed to analyze the prevalence of gastrointestinal parasites in chickens raised in a backyard free-range system in Romania, and the first report of OTUy species in Europe.

## 1. Introduction

The rising need for poultry products in human diets has led to significant expansion in both extensively and intensively managed poultry farming over the past few decades [1]. Consequently, poultry production is increasingly becoming a noteworthy contributor to the national economies of many countries [2].

In 2022, the European Union (EU) generated approximately 13 million tons of poultry meat [3]. According to EUROSTAT, Romania produced a total of 491.22 thousand metric tons, securing the 6th position among EU member states in terms of poultry meat production [4]. Poultry meat constitutes approximately 45 percent of the aggregate animal protein production in Romania, emerging as the predominant meat category in recent years. This marks a noteworthy transition from the preceding decade, during which pork production held the leading position [5]. Furthermore, in accordance with data from the National Institute of Statistics (INS), the production of eggs totaled 6.005 million units, and is anticipated to experience an annual growth rate of 7.63% [6,7].

Even though intensive farms currently dominate the primary production of poultry meat, consumer preferences are gradually moving towards alternative rearing systems, such as free-range and organic [8]. Despite the diminished impact of parasitic diseases in industrial farms attributed to modernization and effective bio-security measures, chickens raised in backyard free-range systems are confronted with a significant threat of parasitic infections. This is particularly due to factors such as unhygienic management practices, litter contamination, and abundance of intermediate hosts [9,10]. Given that numerous farm workers practice extensive chicken raising for personal consumption, they may serve as passive vectors for various diseases, thereby posing a threat to the farm’s biosecurity [11].

Among the parasitic agents, protozoa belonging to the genus *Eimeria* and helminths, including *Ascaridia galli*, *Capillaria* spp., *Heterakis gallinarum*, and *Strongyloides avium*, stand out as the most prevalent [12]. Consequences attributed to parasitic infections include diminished health, welfare, and production efficacy marked by compromised feed conversion ratios, lower growth rates or weight loss, diminished egg production and compromised egg quality, as well as intestinal damage. In severe instances, fatalities may occur [11]. Additionally, an indirect effect is expressed by an increased susceptibility to secondary infectious diseases and a reduction in the host’s immune response [2]. 

*Eimeria* spp. exerts a more pronounced negative impact on the health, welfare and production. The annual estimated worldwide financial burden of coccidiosis in chickens surpassed £10 billion. This cost encompasses expenses related to prevention, treatment, and economic losses [13]. Prophylaxis is achieved through a combination of chemoprophylaxis, vaccination, and dietary supplementation with various plant extracts. However, it still poses a significant threat [14,15,16,17]. 

For a long time, seven species have been identified as infecting chickens, inducing enteritis lesions that result in diarrhea, malabsorption, and hemorrhages [18]. Recently, three cryptic variants, designated as Operational Taxonomic Units (OTUs) X, Y, and Z, have been suggested and assigned as new species, namely *Eimeria lata*, *E. nagambie*, and *E. zaria* [19]. Although initially, the circulation of these species was associated with Australia, Africa, and South America, but recently their presence has been reported in Europe [18].

However, despite the widespread adoption of the free-range system for chicken rearing in Romania and the frequent occurrence of parasitic diseases in this species, there has been no comprehensive study to evaluate their prevalence. Therefore, the current study aims to assess the prevalence of gastrointestinal parasites in chickens raised in the backyard system within the northwestern and central regions of Romania.

## 2. Materials and Methods

### 2.1. Animals, Samples, and Sample Analysis

A total of 290 fecal samples were collected from 145 chicken backyard farms located in Northwest and Central Romania. The number of chickens in a flock varied between households, from 6 to 45, with an average of 20.1 ± 9.5 chickens/flock. The age distribution of the chickens spanned from 1 to 3 years, encompassing both males and females, with the ratio heavily favoring females. The investigation spanned from December 2016 to May 2023. The sampling protocol included 16 localities in 4 counties within the Transylvania region of Romania, as follows: Cluj (Mera, Ceanu Mare, Coruș, Frata), Harghita (Corund, Lupeni, Valea lui Pavel, Ocna de Sus), Mureș (Sovata, Chibed, Sărățeni, Sângeorgiu de Pădure), and Satu Mare (Carei, Urziceni, Foieni, Căpleni) (Figure 1). Fecal samples were collected one time from chickens reared in a backyard agricultural system. Two samples were collected randomly, by hand, from each household. Each fecal sample consisted of 10 pooled droppings, collected from the floor. Subsequently the samples were stored at 4–8 °C until the testing procedure.

Fecal samples were screened by the flotation method. The parasitic elements in positive samples were counted using the McMaster method [20]. Positive samples for *Eimeria* spp. identified through the flotation method were further analyzed using the polymerase chain reaction (PCR) method in order to identify the *Eimeria* species.

### 2.2. PCR

First, DNA extraction was performed from fecal samples (*n* = 18) or concentrated oocysts (*n* = 28) using the commercial kit Isolate Fecal DNA Kit (Bioline, London, England, United Kingdom; Cat. No. BIO-52038) [21].

*Eimeria* species were identified using specific primers for each species (Table 1).

### 2.3. Statistical Analysis

Statistical analysis of the data was conducted using EpiInfoTM 2000 software (version 7.2.0.1, Atlanta, GA, USA). The frequency, prevalence, and its 95% confidence interval (95% CI) of detected species were recorded. The differences in prevalence among identified parasites overall and by the average age of the flock, flock size, and season of sample collection were evaluated using the chi-squared test, or Pearson’s chi-squared test. Based on the age of the chickens, the poultry flocks were divided into flocks with 12–23-month-old chickens and flocks with 24–36-month-old chickens. Depending on the number of chickens in a household, flocks were divided into flocks with 1–10, 11–20, 21–30, and 31–45 chickens, respectively. According to the month of sample collection, two seasons were included in the statistical analysis: winter for samples collected from December to February, and spring for samples collected from March to May. A *p* value of <0.05 was considered statistically significant.

### 2.4. Ethical Statement

The investigation was carried out within backyard farms. The animals under consideration were neither manipulated nor subjected to constraints on their mobility or daily activities. The verbal consent of the flock owners to collect the fecal samples and to publish the results was obtained.

## 3. Results

The overall prevalence of infection with intestinal parasites in free-range chickens was 53.1% (77/145; 95% CI: 45.0–61.0). *Ascaridia galli*/*Heterakis gallinarum* (25.5%) and *Eimeria* spp. (24.8%) were the most prevalent statistically significant (*p* = 0.002) infections, followed by *Capillaria* spp. (23.5%) and digestive strongyles (8.3%) (Table 2). Single infection was recorded in 29.7% (43/145; 95% CI 22.8–37.5), while mixed infection was recorded in 22.8% (33/145; 95% CI 16.7–30.2) of analyzed samples (Table 2). The mean OPG/EPG values, determined through the McMaster method, were as follows: 63,577 for *Eimeria* spp., 157 for *Ascaridia*/*Heterakis*, 362 for *Capillaria* spp., and 1671 for *Strongyle* eggs.

No statistically significant findings were observed in relation to the age of chickens as indicated in Table 3. However, noteworthy statistical significance (*p* = 0.003) was observed for *A. galli*/*H. gallinarum*, a prevalence of 13.1% was recorded in flocks with 11–20 chickens. Also, significant results were recorded (*p* = 0.01) for the mixed infections in the same group of chickens (11.0%), as detailed in Table 4.

Depending on the season in which the samples were collected, namely winter and spring, statistically significant results were obtained for *Capillaria* spp., as well as between the total number of positive samples between the two seasons (Table 5).

Within polyspecific parasitism15.2% (22/145; 95% CI 10.2–21.9) of the flocks were positive for two parasites, while 7.6% (11/145; 95% CI 4.3–13.1) for three parasites. Statistically significant results were recorded within polyspecific parasitism (*p* = 0.03) (Table 6).

The following *Eimeria* species were identified by PCR: *E. acervulina* (41.7%), *E. tenella* (27.8%), *E. praecox* (16.7%), *E. brunetti* (16.7%), OTUy (8.3%), OTUz (8.3%) and *E. mitis* (5.6%). Coinfections involving multiple *Eimeria* spp. were also documented. Statistically significant results were recorded (Table 7). Additionally, one flock tested positive for both OTUz and OTUy species.

## 4. Discussion

The poultry industry holds a crucial position in ensuring food safety and nutritional requirements, emerging as the most rapidly advancing agricultural sub-sector. Anticipated factors influencing sectoral expansion encompass ongoing urbanization trends, population increase, and rising income levels. In 2020, the poultry sector exhibited a market value of $310.7 billion, with projections indicating an ascent to surpass $400 billion by 2025 [26]. Intestinal parasites exhibit a high prevalence among chickens in backyard poultry flocks, and the presence of significant parasite burdens can adversely affect both productivity and economic aspects [27]. However, regular deworming is not commonly practiced in free-range systems, as owners often lack awareness regarding the risks posed by gastrointestinal parasites [28].

A systematic review on the prevalence of gastrointestinal nematodes in chicken published by Shifaw, encompassing nearly 200 studies published over 80 years, revealed that *A. galli*, *H. gallinarum*, and *Capillaria* spp. were the most commonly identified parasites [2]. This aligns with our findings, where these three species exhibited the highest prevalence. 

The pooled prevalence reported by Shifaw for Europe (78.9%) and for the backyard production system (82.6%) surpasses the prevalence observed in our study. This variance could be attributed to ecological, environmental, and climatic factors, including seasonal dynamics, the quantity and accessibility of intermediate hosts, among others. Additionally, variations in diagnostic and sampling procedures, along with diverse host-related factors, may significantly impact the recorded prevalence values [2,29]. The complete absence of cestodes and trematodes species could be elucidated by their more intricate life cycle, in terms of intermediate hosts and environment conditions [2,30].

Polyspecific parasitism was recorded in 11.4% of the samples, strengthening the hypothesis that parasitic infestations usually co-circulate in chickens [31]. This is of significant importance, as the association within parasites with gastrointestinal predilection, such as nematodes and coccidia, may heighten their role in early chick mortality and other productivity losses among adults [31]. 

Similar results were documented in a study conducted in Poland, a prevalence of nearly 35% was recorded for *Eimeria* spp., with *E. acervulina* being the most frequently identified species [28]. 

Thus far, *Eimeria zaria* (OTUz) stands as the sole OTU species reported in Europe. However, we managed to identify two species out of three, namely, OTUy and OTUz. Further studies are imperative, particularly in the intensive sector where coccidiosis remains a significant risk, to evaluate the prevalence of the new OTU species. The heightened risk of the presence of the new species is underscored by the inadequacy of protection conferred by current anticoccidial vaccines [18]. Moreover, given that mechanical vectors constitute the most common way of *Eimeria* oocyst transmission, and considering that many employees in poultry farms own a backyard flock, preventive biosecurity measures should be implemented in order to avoid this route of contamination [32].

To the best of our knowledge, this is the first comprehensive study that aimed to analyze the prevalence of gastrointestinal parasites in chickens raised in a backyard free-range system in Romania. Moreover, this is the first report of OTUy in Europe.

## 5. Conclusions

The widespread distribution of gastrointestinal parasites in chickens raised in the free-range system is most likely explained by poor management practices, including sanitary deficiencies and the absence of deworming programs. Moreover, the scavenging activities of the chickens, a characteristic of this system that enhances contact with excreta, could contribute to this distribution.

Although the prevalence recorded in Romania is relatively lower compared to other European countries, we managed to identify the most prevalent gastrointestinal parasite species.

Considering the upward trend of organic growth systems, there should be an implementation of increased awareness among owners regarding the prevention and availability of treatment methods.

Additional research is necessary to obtain an optimal understanding of the epidemiological status of Romania concerning gastrointestinal parasites in chickens.

## Figures and Tables

**Figure 1 animals-14-00563-f001:**
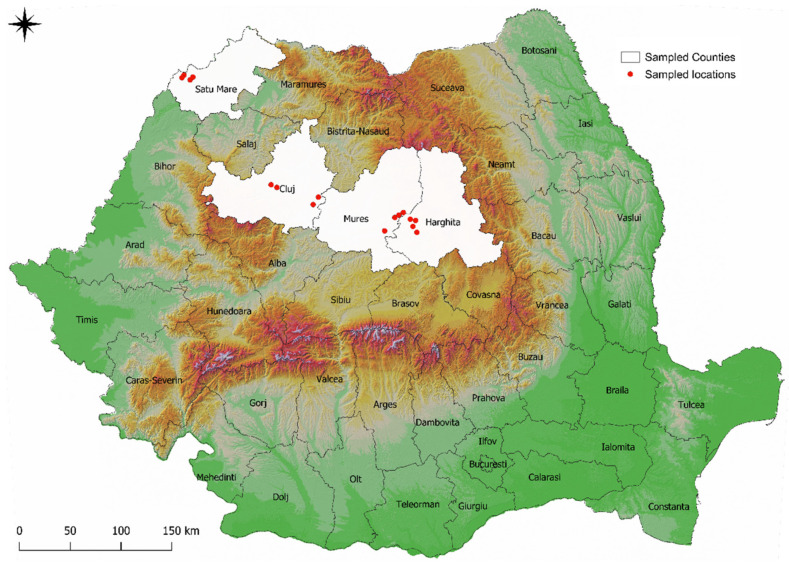
Sampled counties within the Transylvania region of Romania www.qgis.org (accessed on 29 November 2023).

**Table 1 animals-14-00563-t001:** Sequence of specific primers used for identification of *Eimeria* species in backyard chickens in Romania by the PCR method [22,23,24,25].

Species	Primer Sequence 5′ 3′	Annealing Temperature (°C)	Amplicon Size (bp)
*E. acervulina*	F 5′-GGGCTTGGATGATGTTTGCTG-3′ R 5′-GCAATGATGCTTGCACAGTCAGG-3′	65	145
*E. brunetti*	F 5′-CTGGGGCTGCAGCGACAGGG-3′ R 5′-ATCGATGGCCCCATCCCGCAT-3′	58	183
*E. maxima*	F 5′-GTGGGACTGTGGTGATGGGG-3′ R 5′-ACCAGCATGCGCTCACAACCC-3′	65	205
*E. mitis*	F 5′-GTTTATTTCCTGTCGTCGTCTCGC-3′ R 5′-GTATGCAAGAGAGAATCGGGATTCC-3′	65	330
*E. necatrix*	F 5′-AGTATGGGCGTGAGCATGGAG-3′ R 5′-GATCAGTCTCATCATAATTCTCGCG-3′	58	160
*E. praecox*	F 5′-CATCGGAATGGCTTTTTGAAAGCG-3′ R 5′-GCATGCGCTAACAACTCCCCTT-3′	65	215
*E. tenella*	F 5′-AATTTAGTCCATCGCAACCCTTG-3′ R 5′-CGAGCGCTCTGCATACGACA-3′	65	278
*OTUx*	Xf2 5′-GGGTAGAGCCAGGGGTAGAG-3′ Xr2 5′-CGTAGTCCCAAGTGCCAACT-3′	58	1018
*OTUy*	Yf1 5′-CAAGAAGTACACTACCACAGCATG-3′ Yr1 5′-ACTGATTTCAGGTCTAAAACGAAT-3′	56	346
*OTUz*	Zf1 5′-TATAGTTTCTTTTGCGCGTTGC-3′ Zr1 5′-CATATCTCTTTCATGAACGAAAGG-3′	58	147

**Table 2 animals-14-00563-t002:** The frequency, prevalence, and its 95% CI of identified parasitic species by flotation technique (*n* = 145).

Species	Frequency (*n*)	Prevalence (%)	95% CI	*p*-Value
*Eimeria* spp.	36	24.8	18.5–32.5	0.002
*A. galli/H. gallinarum*	37	25.5	19.1–33.2	
*Capillaria* spp.	34	23.5	17.3–31.0	
*Strongyle* egg	12	8.3	4.8–13.9	
Single infection	43	29.7	22.8–37.5	0.251
Mixed infection	33	22.8	16.7–30.2	
Total	77	53.1	45.0–61.0	

Legend: 95% CI—95% confidence interval.

**Table 3 animals-14-00563-t003:** The frequency (prevalence; 95% CI) of identified parasitic species by flotation technique according to the age of the chickens.

Species	12–23 Months (*n* = 69)	24–36 Months (*n* = 76)	*p*-Value
*Eimeria* spp.	14 (9.7; 5.8–15.6)	22 (15.2; 10.2–21.9)	0.182
*A. galli/H. gallinarum*	18 (12.4; 8.0–18.8)	19 (13.1; 8.6–19.6)	0.869
*Capillaria* spp.	17 (11.7; 7.5–18.0)	17 (11.7; 7.5–18.0)	1
*Strongyle* eggs	5 (3.5; 1.5–7.8)	7 (4.8; 2.4–9.6)	0.563
Single infection	16 (11.0; 6.9–17.2)	28 (19.3; 13.7–26.5)	0.07
Mixed infection	16 (11.0; 6.9–17.2)	17 (11.7; 7.5–18.0)	0.861
Total	32 (22.1; 16.1–29.5)	45 (31.0; 24.1–39.1)	0.138

**Table 4 animals-14-00563-t004:** The frequency (prevalence; 95% CI) of identified parasitic species by flotation technique according to the size of the chicken flock.

	1–10 Chickens (*n* = 40)	11–20 Chickens (*n* = 58)	21–30 Chickens (*n* = 30)	31–45 Chickens (*n* = 17)	*p*-Value
*Eimeria* spp.	6 (4.1; 1.9–8.7)	10 (6.9; 3.8–12.2)	10 (6.9; 3.8–12.2)	10 (6.9; 3.8–12.2)	0.957
*A. galli/H. gallinarum*	6 (4.1; 1.9–8.7)	19 (13.1; 8.6–19.6)	5 (3.5; 1.5–7.8)	7 (4.8; 2.4–9.6)	**0.003**
*Capillaria* spp.	7 (4.8; 2.4–9.6)	12 (8.3; 4.8–13.9)	10 (6.9; 3.8–12.2)	5 (3.5; 1.5–7.8)	0.332
*Strongyle* egg	2 (1.4; 0.4–4.9)	5 (3.5; 1.5–7.8)	4 (2.8; 1.1–6.9)	1 (0.7; 0.1–3.8)	0.343
Single infection	13 (9.0; 5.3–14.7)	13 (9.0; 5.3–14.7)	9 (6.2; 3.3–11.4)	9 (6.2; 3.3–11.4)	0.692
Mixed infection	3 (2.1; 0.7–5.9)	16 (11.0; 6.9–17.2)	8 (5.5; 2.8–10.5)	6 (4.1; 1.9–8.7)	**0.010**
Total	16 (11.0; 6.9–17.2)	29 (20.0; 14.3–27.3)	17 (11.7; 7.5)	15 (10.3; 6.4–16.4)	0.08

**Table 5 animals-14-00563-t005:** The frequency (prevalence; 95% CI) of identified parasitic species by flotation technique according to the season of the samples collection.

Species	Winter (*n* = 66)	Spring (*n* = 79)	*p*-Value
*Eimeria* spp.	16 (11.0; 6.9–17.2)	20 (13.8; 9.1–20.4)	0.504
*A. galli/H. gallinarum*	13 (9.0; 5.3–14.7)	24 (16.6; 11.4–23.5)	0.07
*Capillaria* spp.	9 (6.2; 3.3–11.4)	25 (17.2; 12.0–24.2)	**0.006**
*Strongyle* egg	4 (2.8; 1.1–6.9)	8 (5.5; 2.8–10.5)	0.248
Single infection	17 (11.7; 7.5–18.0)	26 (17.9; 12.5–25.0)	0.169
Mixed infection	11 (7.6; 4.3–13.1)	22 (15.2; 10.2–21.9)	0.05
Total	29 (20.0; 14.3–27.3)	48 (33.1; 26.0–41.1.)	**0.03**

**Table 6 animals-14-00563-t006:** Single and mixed parasitic infections. Co-occurrence between species within polyspecific parasitism.

	Frequency (*n*)	Prevalence (%)	95% CI	*p*-Value
Single infection				
*Eimeria* spp.	16	11.0	6.9–17.2	0.121
*Ascaridia/Heterakis*	11	7.6	4.3–13.1	
*Capillaria* spp.	10	6.9	3.8–12.2	
*Strongyle*	5	3.5	1.5–7.8	
Mixed infection				
E + A/H	8	5.5	2.8–10.5	**0.03**
E + C	3	2.1	0.7–5.9	
A/H + C	10	6.9	3.8–12.2	
A/H + S	2	1.4	0.4–4.9	
E + A/H + C	5	3.5	1.5–7.8	
E + C + S	4	2.8	1.1–6.9	
A/H + C + S	1	0.7	0.1–3.8	

Legend: E—*Eimeria*, A/H—*Ascaridia/Heterakis*, C—*Capillaria*, S—*Strongyle* type egg, 95% CI—95% confidence interval.

**Table 7 animals-14-00563-t007:** *Eimeria* species identified by PCR and their coinfections (*n* = 36).

Species	Frequency (*n*)	Prevalence (%)	95% CI	*p*-Value
*E. acervulina*	15	41.7	27.1–57.8	0.002
*E. tenella*	10	27.8	15.9–44.0	
*E. praecox*	6	16.7	7.9–31.9	
*E. brunetti*	6	16.7	7.9–31.9	
OTUy	3	8.3	2.9–21.8	
OTUz	3	8.3	2.9–21.8	
*E. mitis*	2	5.6	1.5–18.1	
A + P	1	2.8	0.5–14.2	0.699
A + T	3	8.3	2.9–21.8	
A + B	3	8.3	2.9–21.8	
M + P	1	2.8	0.5–14.2	
M + T	1	2.8	0.5–14.2	
P + T	3	8.3	2.9–21.8	
A + P + T	2	5.6	1.5–18.1	0.818
A + M + P + T	1	2.8	0.5–14.2	
A + P + T + B	2	5.6	1.5–18.1	

Legend: A—*E. acervulina*, M—*E. mitis*, P—*E. praecox*, T—*E. tenella*, B—*E. brunetti*, 95% CI—95% confidence interval. Mixed infections with OTUy and OTUz are not included in the table.

## Data Availability

All data generated or analyzed during this study are included in this published article. Other datasets used and/or analyzed can be made available by the corresponding author on reasonable request.

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
