# Peer review of "Occurrence of Eimeria spp. and Intestinal Helminths in Free-Range Chickens from Northwest and Central Romania"

_animals, 2024, doi:10.3390/ani14040563_

Round 1

Reviewer 1 Report

Comments and Suggestions for Authors

The manuscript reports the results of investigating 290 fecal samples from 145 small chicken flocks by flotation for parasite stages. Detected coccidia were speciated by PCR. The study is scientifically sound, and the results contribute to the body of knowledge about the epidemiology of parasites. Some clarifications in the manuscript and maybe a re-analysis of the data are necessary.

Most importantly, the results should be aggregated by flock, i.e. how many flocks were positive for a parasite, not how many samples. If the samples were from a single bird, which is unclear, the results per sample may be given additionally, even though two samples per flock is obviously not enough to estimate the in-flock prevalence.

The authors use the chi-square test (lines 111 – 122). The chi-square test analyses the relationship between two categorical variables, and I do not see which two variables those are in each test. If the authors want to compare prevalences, there are more suitable tests for that. The authors should explain better what their null hypothesis was or use another test.

Specific comments

·        Lines 42 – 58 and 146 – 155: This is all about commercial poultry, but the manuscript is about small “backyard” flocks. This part should be replaced with information about small flocks in Romania and maybe in the European Union.

·        Line 86: I would suggest adding the information given in lines 119 – 120 here. Even though this is redundant, it’s also part of Materials and Methods and not only of results.

·        The data should be analyzed if flock size, age / multiple age status and season influenced the prevalence of the parasites. This information about the samples seems to have been collected (lines 86 – 88).

·        Lines 93 – 94: Please describe the system that was followed when collecting the samples. Systematic sample collection also seems to contradict random sample collection. Please clarify. Does “sample” mean one dropping of a single bird, or were these pooled samples? If so, approximately how many droppings were pooled?

·        Reference 24: Please cite the chapter and its authors, not the whole book.

Author Response

  1. The manuscript reports the results of investigating 290 fecal samples from 145 small chicken flocks by flotation for parasite stages. Detected coccidia were speciated by PCR. The study is scientifically sound, and the results contribute to the body of knowledge about the epidemiology of parasites. Some clarifications in the manuscript and maybe a re-analysis of the data are necessary.

  1. Most importantly, the results should be aggregated by flock, i.e. how many flocks were positive for a parasite, not how many samples. If the samples were from a single bird, which is unclear, the results per sample may be given additionally, even though two samples per flock is obviously not enough to estimate the in-flock prevalence.

R: Thank you for your suggestion! We aggregated the results by flock and we provided additional clarification regarding the sample collection (lines 100-103).

  1. The authors use the chi-square test (lines 111 – 122). The chi-square test analyses the relationship between two categorical variables, and I do not see which two variables those are in each test. If the authors want to compare prevalences, there are more suitable tests for that. The authors should explain better what their null hypothesis was or use another test.

R: We used chi-square test when we had 2 variables (as was the case for age and season), and Pearson's chi-squared test when we had more than 2 variables (flock size, and between pasaite sepcies).

Specific comments

  1. Lines 42 – 58 and 146 – 155: This is all about commercial poultry, but the manuscript is about small “backyard” flocks. This part should be replaced with information about small flocks in Romania and maybe in the European Union.

R: So far there are no publication regarding the extensive chicken rearing system in Romania. We added the information about commercial poultry in order to create a context regarding chicken industry in Romania, but also to attempt to establish a premise regarding the role of farm workers as passive vectors for various diseases. We have made additions in this regard.

  1. Line 86: I would suggest adding the information given in lines 119 – 120 here. Even though this is redundant, it’s also part of Materials and Methods and not only of results.

R: Added

  1. The data should be analyzed if flock size, age / multiple age status and season influenced the prevalence of the parasites. This information about the samples seems to have been collected (lines 86 – 88).

R: Added

  1. Lines 93 – 94: Please describe the system that was followed when collecting the samples. Systematic sample collection also seems to contradict random sample collection. Please clarify. Does “sample” mean one dropping of a single bird, or were these pooled samples? If so, approximately how many droppings were pooled?

R: Revised (Fecal samples were collected one time from chickens reared in a backyard agricultural system. Two samples were collected randomly, by hand, from each farm. Each fecal sample consisted in 10 pooled droppings, collected from the floor.)

  1. Reference 24: Please cite the chapter and its authors, not the whole book.

R: Revised (McDougald L.R., 2020. Internal Parasites. Diseases of Poultry (13th ed.), Wiley-Blackwell, New York City, NY, pp. 1117-1147.)

Reviewer 2 Report

Comments and Suggestions for Authors

Dear Correspondence Author,

Happy New Year 2024.

Appreciating your efforts and interesting in publishing in animals Journal. Please find below and in the attached file the comments and suggestions that might enrich the manuscript.

Title: add tract after gastrointestinal and a hyphen (-) between north and west.

Simple summary: Try to highlight the implications such as the economical impact in figuers.

Abstract:

1- productive vs. productivity.

2- Abbreviations (ruled and applications) to be revised.

Introduction and M&M:

1- Required some supported references.

2- Reference number [6] is missing (check).

3- Unify the format for editing numbers.

4- Abbreviations (ruled and applications) to be revised.

5- Title vs. statement. Title did NOT end by full stop.

6- Table (1) should stand alone. Therefore, informative titles are required.

7- Rules for ethical approval in the journal (to be revised).

8- Required details about the number of chickens and obtained samples per each backyard will be favorable and well appreciated.

Results, Discussion, and Conclusion:

1- Informative tables: all abbreviations such as CI, should be in the footnote.

2- Title vs. statement. Delete all full stops from tables' titles.

3- p-values (to be revised, especially in Table (4)).

4- Some supported references are required (revised the attached file).

5- Try to emphasize the economic impact in figures.

Comments on the Quality of English Language

Revise the attached file.

Author Response

Dear Correspondence Author,

Happy New Year 2024.

R: Thank you very much! Wish you a great New Year!

Appreciating your efforts and interesting in publishing in animals Journal. Please find below and in the attached file the comments and suggestions that might enrich the manuscript.

Title: add tract after gastrointestinal and a hyphen (-) between north and west.

R: The title has been changed, at the editor's suggestion.

Simple summary: Try to highlight the implications such as the economic impact in figures.

R: We added some information regarding the economic impact.

Abstract:

1- productive vs. productivity.

R: Revised

2- Abbreviations (ruled and applications) to be revised.

R: Revised

Introduction and M&M:

1- Required some supported references.

R: Added

2- Reference number [6] is missing (check).

R: Revised

3- Unify the format for editing numbers.

R: Revised

4- Abbreviations (ruled and applications) to be revised.

R: Revised

5- Title vs. statement. Title did NOT end by full stop.

R: Revised

6- Table (1) should stand alone. Therefore, informative titles are required.

R: Revised

7- Rules for ethical approval in the journal (to be revised).

R: The animals under consideration were neither manipulated nor subjected to constraints on their mobility or daily activities. The fecal samples were collected from the floor, in the absence of the chickens. According to the Directive 2010/62/EU of the Parliament and of the Council of 22 September 2010 on the protection of animals used for scientific, Chapter 1 – General provisions, Article 1 - Subject matter and scope, point 5 - This Directive shall not apply to the following: letter f, “practices not likely to cause pain, suffering, distress or lasting harm equivalent to, or higher than, that caused by the introduction of a needle in accordance with good veterinary practice”. Furthermore, Law number 43/2014 of Romania, which regulates the protection of animals used for scientific purposes, specifies the same aspects as European Directive 63/2010 (https://eur-lex.europa.eu/LexUriServ/LexUriServ.do?uri=OJ:L:2010:276:0033:0079:en:PDF, https://lege5.ro/gratuit/gm4tomrtge/legea-nr-43-2014-privind-protectia-animalelor-utilizate-in-scopuri-stiintifice)

8- Required details about the number of chickens and obtained samples per each backyard will be favorable and well appreciated.

R: Revised

“Depending on the size of the flock, the households have been divided into 4 groups (gr) as follows: gr 1 = 1-10 chickens; gr 2 = 11-20 chickens; gr 3 = 21-30 chickens; gr 4 = 31-45 chickens.”

“Fecal samples were collected one time from chickens reared in a backyard agricultural system. Two samples were collected randomly, by hand, from each farm. Each fecal sample consisted in 10 pooled droppings, collected from the floor.). Unfortunately, data on the number of chickens were not recorded in all households."

Results, Discussion, and Conclusion:

1-           Informative tables: all abbreviations such as CI, should be in the footnote.

R: Revised

2- Title vs. statement. Delete all full stops from tables' titles.

R: Revised

3- p-values (to be revised, especially in Table (4)).

R: Revised

4- Some supported references are required (revised the attached file).

R: Added

5- Try to emphasize the economic impact in figures.

R: The economic impact was not the focus of this article; we aimed only to emphasize the importance of gastrointestinal parasites in chicken rearing

Round 2

Reviewer 1 Report

Comments and Suggestions for Authors

The authors addressed all my comments adequately.

Reviewer 2 Report

Comments and Suggestions for Authors

Appreciating your efforts.

Best wishes,

Reviwer